# Cardiac Adverse Events after Vaccination—A Systematic Review

**DOI:** 10.3390/vaccines10050700

**Published:** 2022-04-29

**Authors:** Kanak Parmar, Sai Subramanyam, Gaspar Del Rio-Pertuz, Pooja Sethi, Erwin Argueta-Sosa

**Affiliations:** 1Department of Internal Medicine, Texas Tech University Health Sciences Center, Lubbock, TX 79430, USA; gaspar.del-rio-pertuz@ttuhsc.edu; 2School of Medicine, Vydehi Institute of Medical Sciences and Research Centre, Bangalore 560048, India; saiptp999@gmail.com; 3Department of Cardiovascular Medicine, Texas Tech University Health Sciences Center, Lubbock, TX 79430, USA; pooja.sethi@ttuhsc.edu (P.S.); erwinargueta.sosa@ttuhsc.edu (E.A.-S.)

**Keywords:** myocarditis, vaccination, pericarditis, myocardial infarction, adverse events

## Abstract

The Vaccine Adverse Event Reporting System database has been used to report adverse events following several vaccines. We studied the patient population predisposed to such reactions and how these reactions differ with respect to the vaccine type. We searched the electronic databases PubMed, EMBASE, and Scopus up to 9 July 2021 for any study describing cardiac adverse events attributed to the vaccination. A total of 56 studies met the criteria comprising 340 patients. There were 20 studies describing cardiac adverse events following smallpox vaccination, 11 studies describing adverse events after influenza vaccination, and 18 studies describing adverse events after COVID-19 vaccination. There was a total of six studies describing cardiac adverse events after the pneumococcal vaccine, tetanus toxoid, cholera vaccine, and rabies vaccine. Adverse events following influenza vaccination occurred more commonly in older females within an average duration of four days from vaccination. Pericardial involvement was the most reported adverse event. Adverse events following COVID-19 vaccination happened at a mean age of 42.7 years, more commonly in males, and mostly after a second dose. Adverse events following smallpox vaccination occurred more commonly in younger males, with an average onset of symptoms from vaccination around 16.6 days. Adverse events were mostly myopericarditis; however, the acute coronary syndrome has been reported with some vaccines.

## 1. Introduction

Vaccination has remained an integral part of primary care medicine for preventing common and life-threatening diseases for decades. Vaccination has been associated with minor injection site reactions, fever, fatigue, and lymphadenopathy; however, serious neurological and cardiac adverse events (AEs) have been known to occur [1]. The Vaccine Adverse Event Reporting System (VAERS), a passive surveillance database, provides information on reports of AEs after vaccination with approved vaccines in the United States (2). Through this passive reporting, the Centers for Disease Control and Prevention (CDC) and the US Food and Drug Administration (FDA) conduct post-licensure vaccine safety monitoring [2]. A study on the VAERS database from 1990 to 2018 showed 0.1% (708) myopericarditis cases out of the 620,195 reports of possible adverse events to VAERS [3]. At the end of 2019, a novel coronavirus now known as severe acute respiratory syndrome coronavirus 2 (SARS-CoV-2) was identified as the cause of pneumonia cases in Wuhan, China [4]. It rapidly spread, resulting in a global pandemic affecting 400 million people worldwide and has taken approximately 6 million lives till now. In order to fight this infection, there was an emergent authorization of several vaccines by the World Health Organization all over the world. Two of them are Coronavirus disease (COVID-19) mRNA vaccines: BNT162b2 (Pfizer-BioNTech COVID-19 vaccine) and mRNA-1273 (Moderna COVID-19 vaccine), three are adenoviral vector vaccines: Ad26.COV2.S (Janssen COVID-19 vaccine, also referred to as the Johnson & Johnson vaccine) and AZD1222(Oxford/AstraZeneca), Covishield (Oxford/AstraZeneca), and lastly, three are inactivated vaccines: Covaxin (Bharat Biotech), BBIBP-CorV (Sinopharm) and CoronaVac (Sinovac) [5]. Soon after, various reports of adverse events (AEs) from the COVID-19 vaccines emerged [6,7,8,9]. The aim of our review is to investigate the patient characteristics and provide an overview of the management for patients that develop AEs after vaccination.

## 2. Methods

### 2.1. Design

We conducted a systematic review with a priori selection and outcome criteria according to “Preferred Reporting Items for Systematic Reviews and Meta-Analyses” guidelines. We registered the study protocol in Prospero with CRD42021267467. We searched the electronic databases PubMed, EMBASE, and Scopus. The search included articles published from inception up to 9 July 2021. The MeSH (Medical Subject Headings) search terms were “myocarditis” or “myocardium” or “pericardial” or “pericardium” or “acute myocardial injury” OR “acute coronary syndrome” OR “heart failure” OR “arrhythmia” OR “troponin” OR “ischemia” OR “acute myocardial infarction” OR “coronary events” OR “creatine kinase” OR “heart” AND “vaccine”.

We excluded studies involving patients below 18 years of age. We included a study if it described cardiac AEs associated with the vaccination. We excluded any animal studies. Our primary outcome was to see the cardiac AEs reported to any vaccination.

### 2.2. Study Selection

Two authors (KP and GDRP) independently reviewed the retrieved abstracts and assessed eligibility. The full-text review was conducted when either of the reviewers of the abstracts felt that the citations might meet inclusion criteria. Disagreement was resolved by the engagement of a third author (SS).

### 2.3. Data Extraction

Data from the included studies were independently extracted by three authors (KP, GDRP, SS). We extracted the following data: age, gender, comorbidities, vaccine type, vaccine dose, number of days after which the event happened, inflammatory markers, imaging, results of an endomyocardial biopsy, and treatment instituted.

## 3. Results

Overall, 7327 studies were found on the PubMed, Embase, and Scopus databases (Figure 1). After removing duplicates, 6095 studies were screened by title and abstract for any cardiac AEs after vaccination. A total of 267 studies qualified and were opened for a full-text review. A final selection of 56 studies was selected for adverse cardiac events after vaccination, with a total of 340 patients. Eleven studies described cardiac AEs following influenza vaccination [10,11,12,13,14,15,16,17,18,19,20], and 18 studies described cardiac AEs after COVID-19 vaccination [6,7,8,9,21,22,23,24,25,26,27,28,29,30,31,32,33,34,35]. There were 20 studies reporting cardiac AEs after smallpox vaccination [36,37,38,39,40,41,42,43,44,45,46,47,48,49,50,51,52,53,54,55]. Among other vaccines, there were two reports of cardiac AEs after pneumococcal vaccination [56,57], three reports of cardiac AEs after tetanus toxoid [58,59,60], one report following cholera vaccination [61], and one report following rabies vaccine [62].

The median age for developing cardiac AEs following vaccination was 43.79 ± 21.2 years. Most of the patients described were males (84%) and rest were females (Table 1).

A total of 34 patients were included in cardiac AEs following influenza vaccination, with the mean age of patients being 68.55 ± 18.23 years. Fifty-five percent of cases described were female. Myocarditis/pericarditis/myopericarditis developed in 29 patients, and Takotsubo cardiomyopathy was described in two cases. The average time of symptom onset from vaccination was 4.7 ± 4 days. Anti-inflammatory treatment was used in 66% of patients, and one patient required an extracorporeal membrane oxygenator. Steroids were used on one patient. All patients recovered (Table 2).

A total of 67 patients developed cardiac AEs following COVID-19 vaccination. Overall, 35 cases reported AEs after BNT162b2 (Pfizer), 26 cases developed cardiac AEs after mRNA-1273, four cases developed cardiac AEs after AZD1222, and two cases developed cardiac AEs after receiving CoronaVac and JNJ-78436735. The average age for these patients was 42.7 years (SD = 19.6), with 59 (88%) patients being male gender. Most patients (69.6%) developed a reaction after the second dose. Myocarditis/pericarditis/myopericarditis developed in 35 patients. Acute coronary syndrome (ACS) was described in six patients. There was one case of myocardial infarction (MI) with non-obstructive coronary arteries (MINOCA) and one stress-induced cardiomyopathy. The average time of symptom onset from vaccination was 2.34 days (SD = 1.83 days). Moreover, 15 cases received anti-inflammatory treatment, and the same number of patients received colchicine. All patients recovered except one (Table 3).

A total of 232 developed adverse events following smallpox vaccination with a mean age of 29.48 ± 8.9 years. Most of these AEs (82.75%) were described in male patients. Most patients (212) developed myocarditis/myopericarditis or pericarditis. Twenty-six patients had ACS, and there were two cases of arrhythmias. The average duration of symptom onset from the time of vaccination was 16.6 days (SD = 14). For treatment, nine cases described the use of NSAIDs, and three patients received colchicine. Most patients developed AE following the first dose, but 30 patients developed a reaction after the second dose. There were three cases with mortality. Endomyocardial biopsy (EMB)was performed in eight cases, and four cases had eosinophilic infiltration (Table 4).

Cardiac AEs were also described with the pneumococcal vaccine, mostly in the elderly age group. Myopericarditis was also reported with tetanus toxoid in three males with an average age of 25 years. All three patients recovered, and a biopsy in one case showed eosinophilic infiltration. There was also one case of ACS reported after a cholera vaccination and myocarditis after a rabies vaccination (Table 5).

## 4. Discussion

Myocarditis is characterized by inflammation of the heart, and in resource-abundant countries, viral infections are the most frequently presumed cause of myocarditis [63]. Often pericarditis and myocarditis are observed in tandem; hence the term myopericarditis being recognized by the European Society of Cardiology (ESC). The annual incidence of myocarditis in the United States is estimated to be 1 to 10 per 100,000 of the population [44]. Vaccine-associated myocarditis is a rare event that was recognized as an adverse event after the mass revaccination against the smallpox virus began in military personnel. The new guidelines by ESC have now recognized inflammatory cardiomyopathy into four groups based on EMB results: inflammation-negative, virus-negative; inflammation-positive, virus-negative; inflammation-negative, virus-positive; and inflammation-positive, virus-positive [64]. Vaccine-associated cardiomyopathy usually falls into the virus-negative category. Characteristics of predisposed patients and treatment strategies remain to be defined.

The incidence of myopericarditis rose from 0.08 per 1000 to 0.11 per 1000 after the resumption of smallpox vaccination in 2002 [37]. Recent vaccination against the deadly COVID-19 virus has raised similar concerns of myopericarditis with an absolute rate of 1.7 per 1,000,000 vaccinated individuals as described by Husby et al. in a Denmark-based cohort study. However, these rates are much lower than the incidence rate described for viral myocarditis (10 to 22 per 100,000 individuals) [65]. Thus, there should be no vaccine hesitancy based on these cardiac AEs; however, physicians should be cautious about the development of these AEs, and care should not be delayed if suspicion arises. 

Su et al.’s study on the VAERS database from 1990 to 2018 showed that most patients developing myopericarditis from vaccination were 19–49 years old, and 90% of this age group were male with symptom onset 8–14 days after vaccination [3]. The vaccines frequently associated with these AEs in order of frequency were smallpox, anthrax, typhoid, and inactivated influenza. Cardiac AEs from hepatitis B, zoster vaccine, hepatitis A, varicella, hemophilus, influenza, polio, and pneumococcal vaccine were also reported. We tried to study how these reactions differ with respect to the type of vaccine.

AEs following smallpox vaccination are the most studied among all vaccines. A prospective study by Engler et al. in military personnel saw an increased incidence of new-onset cardiac symptoms following smallpox and trivalent inactivated influenza with a relative risk difference of 16.11 between vaccinated and non-vaccinated populations [37]. Thus, vaccinia-associated inflammatory disease was defined as any cardiac inflammatory syndrome occurring within 30 days of vaccination without another identifiable cause [66]. Reif et al. described that the hyperactivation of inflammatory response from the variola virus in the smallpox vaccine is responsible for these AEs following smallpox vaccination. The study identified increased monocyte recruitment followed by upregulation of intercellular adhesion molecule 1 in patients developing adverse events. The activated macrophages then produce cytokine interleukin-10 (IL-10), which along with certain genotypes of IL-4, leads to increased production of granulocyte stimulating factor-3, a cytokine produced by activated T cells, macrophages, and endothelial cells to increase production of neutrophils for inflammatory reactions [67].

Influenza vaccination has been associated with a decrease in all-cause mortality in heart failure patients [68]. A literature search in 2017 described seven cases of pericarditis in patients above 60 years of age following influenza vaccination, as was seen in our analysis [12]. In a case series of 84 pericarditis cases by Zanettini et al., 23 cases were thought to be due to influenza vaccination in elderly females [18]. The mean time of symptom onset was seven days, as was seen in our analysis [12]. The mechanisms of systemic immunologic reactivity for pericarditis remain to be proven because of its rarity. Following influenza vaccination, there is a systemic inflammatory reaction, and it is postulated that the AEs, particularly TTC, may be due to this increased sympathetic discharge [15].

The first reports of AEs following COVID-19 vaccination came from Israel’s Health Ministry, which reported heart inflammation in cases who received the Pfizer vaccine [69]. Soon after, a CDC advisory committee on immunization practices identified a likely association between the COVID-19 mRNA vaccines and cases of myocarditis and pericarditis [70]. Based on data from the VAERS, the CDC has estimated that the incidence of myocarditis after COVID-19 vaccination is 0.48 cases per 100,000 overall and 1.2 cases per 100,000 among vaccine recipients between the ages of 18 and 29 years. However, based on the study by Witberg et al. on the Israel database incidence of myocarditis, it was estimated that there were 2.13 cases per 100,000 vaccinated persons in the 42 days after the first vaccine dose [71]. It is important to consider these case reports within the broader context of the COVID-19 pandemic, which has caused tremendous morbidity and mortality throughout the world.

COVID-19 vaccination, especially AstraZeneca, was also found to be temporally related to thrombosis, but causality could not be proved; hence the vaccine was suspended [26,72]. Myocardial infarction (MI), in particular, is one of the most dreaded cardiac complications, as was seen in our included studies. The initial data from clinical trials by FDA briefing documents demonstrated that the incidence of MI was 0.02% and 0.03%, respectively, in the vaccine group. Later, a study on an elderly age group, comparing vaccinated and unvaccinated patients, did not show any significant increase in any cardiovascular events such as stroke or pulmonary embolisms [73].

As described earlier, these cardiac AEs are rare and different mechanisms have been proposed for these reactions, including molecular mimicry between immunogens in vaccines and human cells [74,75]. There is also the possibility of interaction between the encoded viral spike protein, antibodies generated by the host, and a yet undetermined cardiac protein in susceptible hosts [75]. Although these mechanisms for AEs are still enigmatic, a preponderance for male gender and younger age was observed in the VAERS database and our review [76]. The association of myocarditis with male sex and younger age could be attributed to sex hormones which may account for a more intense inflammatory response [77]. As suggested by experimental studies on myocarditis in mice, testosterone may be implicated in the inhibition of anti-inflammatory cells and the stimulation of immune responses by mRNA vaccine [78]. The presentation of symptoms within approximately two days of receiving a second dose (68% of patients) of mRNA vaccination also suggests an immune-mediated reaction in the host. Furthermore, cases of Type 1 Kounis syndrome have been described after inactivated COVID-19 vaccine, indicating an allergic reaction to a vaccine component [32].

According to 2012 ESC, cardiac magnetic resonance imaging (CMR) is the noninvasive gold standard method for the diagnosis of myocarditis [65]. CMR findings, including regional dysfunction, late gadolinium enhancement, and elevated native T1 and T2, have been used in many of these cases for myocarditis diagnosis following COVID-19 vaccination [7,9,22]. EMB, which is the gold standard, was performed on five patients with myocarditis after smallpox vaccination and in one patient after COVID-19 vaccination [65]. Three of them showed eosinophilic infiltrate. Yamamoto et al. also described biopsy-proven eosinophilic myocarditis following tetanus toxoid, which responded to high-dose corticosteroid treatment [59].

Specific guidelines for the management of vaccinia-associated myopericarditis have been outlined by the Department of Defense Vaccine Healthcare Center, and symptomatic patients should receive treatment with analgesics and/or NSAIDs as a first-line treatment [66]. Most cases in our study received colchicine and ibuprofen. In patients with persistent symptoms, steroids were advised [66]. The ESC guidelines advise immunosuppressive therapy for virus-negative myocarditis; however, the data are still unclear [79]. It is recommended that in patients refractory to standard therapy with no contraindications, treatment must be tailored on an individual basis [79]. Steroids were used for many patients in this review, which led to improvement.

## 5. Limitations

The review could not include all studies from the VAERS database. It was limited to full-text articles describing the patient and the cardiac AEs. The review also mostly comprised of case reports, case series, and retrospective studies with few prospective studies. Therefore, there is a potential risk of bias, and the results should be interpreted with some caution. Given the rarity of these events and the retrospective nature of the events, it is not possible to estimate the relative risk of these AEs. Lastly, most of the studies did not provide all the required information, particularly about the results of cardiac testing and management.

## 6. Conclusions

Although vaccination remains a pivotal pillar of our healthcare to fight against some of the deadliest infections, understanding vaccine-associated cardiac adverse events will improve our healthcare delivery to this subpopulation. The incidence of cardiac AEs from vaccination remains much lower than cardiac AEs from other causes, but providers should be cautious of these AEs after vaccination.

## Figures and Tables

**Figure 1 vaccines-10-00700-f001:**
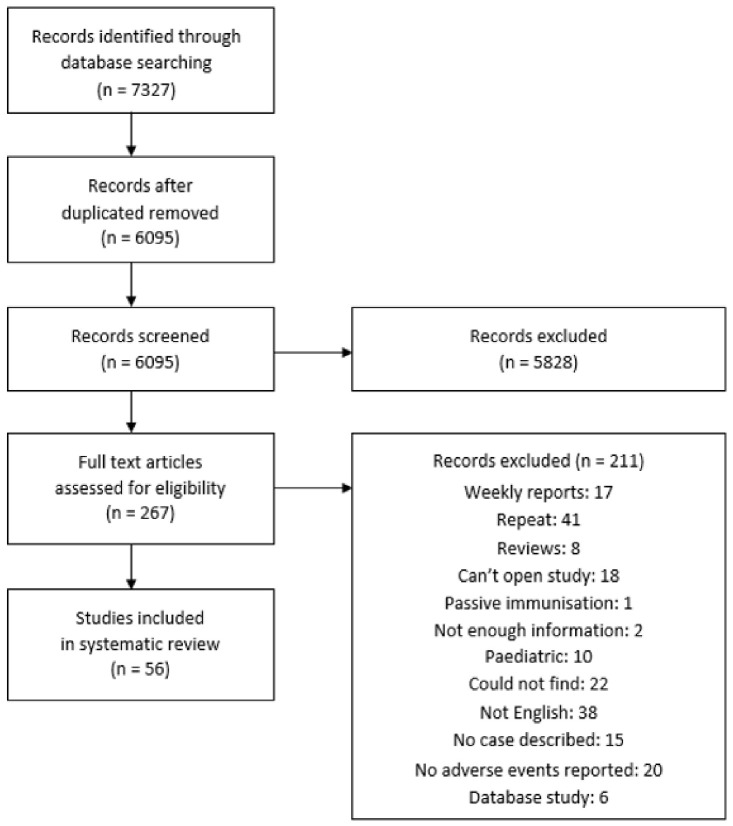
Preferred Reporting Items for Systematic Reviews and Meta-Analyses (PRISMA) Flow diagram.

**Table 1 vaccines-10-00700-t001:** Summary of Vaccine Related Cardiac Adverse Events.

Vaccine Type	No. of Studies	No. of Patients	Average Time of Symptom Onset from Vaccination (Days)	Mean Age (±SD)	Gender	Types of Events	Outcome
Influenza vaccine	11	34	4.77	68.55 ± 18.23	Males: 15 Females: 19	Myocarditis: 2 Pericarditis: 27 Pericardial effusion: 2 Takotsubo: 2 Kounis syndrome: 1	Recovery in all patients
COVID-19 vaccines. Subtype: J&J: 1 Pfizer: 35 Moderna: 26 Covishield: 4 CoronaVac: 1	18	67	2.34	42.733 ± 19.6	Males: 59 Females: 8	Myocarditis: 55 MI: 6 Acute myocardial injury: 1 Type 1 Kounis syndrome: 1 MINOCA: 1 Pericarditis: 2	1 death. Recovery in all other patients.
Smallpox vaccine	20	232	16.68	29.48 ± 8.97	Males: 192 Females: 29 NR: 11	Myocarditis: 75 Pericarditis: 7 Myopericarditis: 121 Ischemic events: 26 DCM: 2 Trigeminy: 1 Ventricular tachycardia: 1	1 death. Recovery in all other patients.
Pneumococcal Vaccine	2	2	18	73.00 ± 2.8	Females: 2	Myocarditis: 1, Pericardial effusion: 1	Recovery in all patients
Tetanus toxoid	3	3	3	25.66 ± 6.8	Males: 3	Myocarditis: 2 Myopericarditis: 1	Recovery in all patients
Cholera	1	1	6	40	Male: 1	Myocardial infarction: 1	Recovery
Rabies	1	1	NR	58	Male: 1	Myocarditis: 1	NR

Abbreviations: NR, Not reported, No, number; SD, standard deviation; J&J: Johnson and Johnson vaccine; DCM, dilated cardiomyopathy.

**Table 2 vaccines-10-00700-t002:** Adverse events after Influenza vaccination.

Author, Year	Country	Type of Study	Number of Cases	Median Age	Gender	Patient Comorbidities	Event	Dose after Which Event Happened	Days after Vaccination Symptoms	Treatment	Outcome	Was EMB Done? If Yes, Findings
Chuen-Der Kao et al., 2003 [10]	Taiwan	Case report	1	68	F	NR	Pericarditis	1	14	Plasmapheresis	R	N
Y J Kim et al., 2017 [11]	Korea	Case report	1	27	F	NR	Myocarditis	1	3	ECMO	R	N
De Meester et al., 2000 [13]	Belgium	Case Report	2	75, 40	M	1: CKD, DM, smoker, 2: smoker, hyperlipidemia	1: pericardial effusion on echo, 2: pericarditis	1	4	NSAIDs	R	N
Santoro et al., 2013 [14]	Italy	Case report	1	78	M	NR	TTC	1	3	Furosemide, ramipril, and bisoprolol	R	N
Singh et al., 2013 [15]	Australia	Case report	1	86	F	NR	TTC	1	1	NA	R	N
Streifler et al., 1981 [16]	Israel	Case report	1	61	M	NR	Pericarditis	1	7	NSAIDs	R	N
Cheng et al., 2016 [17]	Canada	Case report	1	65	M	None	Myocarditis	1	5	NA	R	N
Mei et al.,2018 [12]	Italy	Case report	1	87	M	COPD, thyroidectomy, MI, total AV block with SSS	Pericardial effusion	2	1st event: 5–7 days; 2nd event: few days; 3rd event: relapse: 8 months later	Prednisone pericardiocentesis	R	N
Garcia et al., 2013 [20]	Spain	Case report	1	69	M	IHD	Kounis syndrome	1	15 min	Nitroglycerin	R	N
Zanettini et al., 2003 [18]	Brazil	Case series	23	<40: 1; 40–59: 3; ≥60: 19	16 F, 7 M	Obesity: 10, smokers: 10, HTN: 3, IHD: 2, HLD: 6, mitral valvopathy: 13, HTN cardiomyopathy: 8, arrythmias: 7, COPD: 3. MI: 1, hypothyroidism: 1	Pericarditis	1	NA	19 patients: NSAIDs	R	N
Godreuil et al., 1983 [19]	France	Case report	1	87	M	NR	Hemorrhagic pericarditis	1	NA	Surgical pericardiectomy, colchicine	R	N

MI, myocardial infarction; TTC, takotsubo cardiomyopathy; IHD, ischemic heart disease; HLD, hyperlipidemia; NSAID nonsteroidal anti-inflammatory drug; HTN, hypertension; COPD, chronic obstructive pulmonary disease; CKD, chronic kidney disease; DM, diabetes mellitus, ECMO, extracorporeal membrane oxygenation; R, Recovery; D, Death; SSS, sick sinus syndrome; EMB, endomyocardial biopsy;NR, Not reported.

**Table 3 vaccines-10-00700-t003:** Adverse events after COVID-19 vaccination.

Author, Year	Country	Type of Study	Type of Vaccine	Number of Cases	Median Age	Gender	Patient Comorbidities	Event	Dose after Which Event Happened	Days after Vaccination Symptoms	Treatment	Outcome	Was EMB Done? If Yes, Findings
Rosner et al., 2021 [6]	USA	Case series	J&J: 1, Pfizer: 5, Moderna: 1	7	28, 39, 39, 24, 19, 20, 23	7 M	NR	Myocarditis	1: 2 pts, 2: 5 pts	3 to 7 days	NA	R	1/7: One patient underwent endomyocardial biopsy without pathological evidence of myocarditis
Kim et al., 2021 [7]	USA	Retrospective	Pfizer: 2, Moderna: 2	4	36, 23, 70, 24	3 M, 1 F	Pt 3: HTN, HLD, cigarette smoking; Others: None	Myocarditis	2: 4 pts	3, 5, 1, 2	1: Colchicine, NSAIDs; 2: Colchicine, corticosteriods; 3: Nothing; 4: Colchicine, NSAIDs	R	N
Larson et al., 2021 [8]		Case series	Moderna: 3, Pfizer: 5	8	22, 31, 40, 56, 26, 35, 21, 22	8 M	NR	Myocarditis	1: 7 pts, 2: 1 pts	2–4 days	5 pts were treated with NSAIDs, colchicine, and prednisone	R	N
Mansour et al., 2021 [9]	USA	Case series	Moderna: 2	2	25, 21	1 M, 1 F	NR	Myocarditis	2: 2 pts	1: same day; 2: the next day	Pt. 2 metoprolol	R	N
Lee et al., 2021 [23]	Singapore	Case Series	Pfizer: 3	3	70, 44, 73	3 F	1: Type 2 DM, HTN, HLD, prior CVA; 2: Mitral valve prolapse and mild mitral regurgitation; 3: HTN	1: STEMI with 100% LCX occlusion; 2: Stress-induced cardiomyopathy; 3: MINOCA	1: 3 pts	1: same day; 2: same day; 3: same day	Pt. 1: PCI	R	N
Habib et al., 2021 [24]	Qatar	Case report	Pfizer: 1	1	37	M	HTN	Myocarditis	2: 1	3 days	LHC	R	N
Abu Mouch et al., 2021 [25]	Israel	Case series	Pfizer: 6	6	1: 24, 2: 20, 3: 29, 4: 45, 5: 16, 6: 17	6 M	NR	Myocarditis	1: 1 pt, 2: 5 pts	1: 72 h, 2: 24 h, 3: 48 h, 4: 16 days, 5: 24 h, 6: 72 h	NSAIDs and colchicine;	R	N
Chatterjee et al., 2021 [26]	India	Case report	Covishield: 1	1	63	M	NR	Myocardial infarction	2	48 h	Thrombolysis	R	N
Deb et al., 2021 [27]	USA	Case report	Moderna: 1	1	67	M	HTN, DM 2, HLD, CAD s/p stents and CABG	Acute myocardial injury	2	6 h	Diuretics	R	N
Albert et al., 2021 [28]	USA	Case report	Moderna: 1	1	24	M	NR	Myocarditis	2	24 h	Beta-blocker	R	N
Ammirati et al., 2021 [31]	USA	Case report	Pfizer: 1	1	56	M	NR	Acute myocarditis	2	3 days	NA	R	N
Özdemir et al., 2021 [34]	Turkey	Case report	CoronaVac: 1	1	41	F	NR	Type 1 Kounis syndrome	1	15 min			N
D’Angelo et al., 2021 [29]	Italy	Case report	Pfizer: 1	1	30	M	NR	Myocarditis	2	72 h			N
Tajstra et al., 2021 [32]	Poland	Case report	Pfizer: 1	1	86	M	Prostate cancer, Paroxysmal A.fib.	Acute MI	1	30 min			N
Montgomery et al., 2021 [22]	USA	Case series	Pfizer: 7, Moderna:16	23	Median age: 25	23 M	NR	Myocarditis	1: 3 pt; 2: 20 pts	4 days			N
Watad et al., 2021 [35]	Israel	Case series	Pfizer: 2	2	48, 66	1 M, 1 F	Pt 1: HTN, HLD, HCM; Pt 2: Idiopathic pericarditis, anemia, DVT, spontaneous abortion in 1st trimester	1: pericarditis; 2: pericarditis	1: 2 pts; 1 pt had relapse after 2nd dose	1: after 4 days for both events; 2: 2 days			N
Watkins et al., 2021 [30]	USA	Case report	Pfizer: 1	1	20	M	NR	Myocarditis	2	2 days			N
Srinivasan et al., 2021 [33]	India	Case series	Covishield: 3	3	46, 48, 75	2 M, 1 F	1: DM, HTN; 2: DM, HTN, CAD; 3: NA	1: TVD, MI RCA occlusion, 2: LAD occlusion MI; 3: TVD, occlusion of postero-lateral branch of RCA	2: 2 pts, 1: 1 pt	1: 12 days; 2: 6 days; 3: 1 day			N

TTC, takotsubo cardiomyopathy; IHD, Ischemic heart disease; HLD, hyperlipidemia; NSAID, nonsteroidal anti-inflammatory drug; HTN, hypertension; COPD, chronic obstructive pulmonary disease; CKD, chronic kidney disease; DM, diabetes mellitus, ECMO, extracorporeal membrane oxygenation; STEMI, ST elevation myocardial infarction; CAD, coronary artery disease; RCA, right coronary artery; LHC: left heart catheterization; R, Recovery; D, Death; EMB, endomyocardial biopsy; NR, not reported, A fib, atrial fibrillation.

**Table 4 vaccines-10-00700-t004:** Adverse events after Smallpox vaccination.

Author, Year	Country	Type of Study	Number of Cases	Mean Age (Years)	Gender	Patient Comorbidities	Event	Dose after Which Event Happened	Days after Vaccination Symptoms	Treatment	Outcome	EMB(Y/N). If Y, findings
Whitman et al., 2003 [36]	USA	Case Report	1	29	F	NR	Frequent episodes of trigeminy associated with symptoms of palpitations	1	10	Beta blocker	R	N
Lin et al., 2013 [55]	USA	Retrospective	11	NA	NA	NR	4 pericarditis, 7 myopericarditis	1	21	Supportive	R	N
Engler et al., 2015 [37]	USA	Prospective Cohort	5	NA	4 M, 1 F	NR	4 pts myocarditis; 1 pt pericarditis	1	NA	Supportive	R	N
						NR	All pts: subclinical myocarditis	1	NA	Supportive	R	N
Eckart et al., 2004 [38]	USA	Systematic surveillance	67	26.6	66 M, 1 F	NR	67 myocarditis	1	10.4	Supportive	1 fatality: 33 days after multiple vaccinations	Y,1 fatality. Autopsy: eosinophilic epicardial inflammation;
Sarkisian et al., 2019 [39]	USA	Case Series	6	23	6 M	NR	2 cases: myocarditis; 4 cases: myopericarditis	1	10.5	Pt1. colchicine, Pt2. NSAIDs, colchicine, Pt.3. NSAIDs, Pt4. NA, Pt5. NSAIDs, colcichine, Pt6. NSAIDs	R	N
Saurina et al., 2003 [40]	USA	Case series	1	20	M	NR	Myocarditis	1	8	NSAIDs	R	N
Sharma et al., 2011 [41]	USA	Case Series	2	34	2 M	NR	1 case myocarditis, 1 case myopericarditis	1	12	NSAIDs	R	N
Sniadack et al., 2008 [42]	USA	Cohort prospective	33		13 M, 20 F	NR	10 cases: ischemic event, 2 cases: DCM and 21 cases: myopericarditis	1: 4 pts; Revaccinee: 29 pts;	11	Supportive	R	N
Keinath et al., 2018 [43]	USA	Case study	1	36	M	NR	1 myocarditis	1	70	Cardioversion, ICD placement, GDMT for HF	R	N
Halsell et al., 2003 [44]	USA	Case series	18	26.5	18 M	NR	18 cases of myopericarditis	1	10.5	Supportive	R	Y, eosinophilic infiltration
Mathews et al., 1974 [45]	England	Case report	1	25	M	NR	1 myopericarditis	1	14	Hydrocortisone, Prednisone Digoxin and furosemide	R	N
Eckart et al., 2005 [46]	USA	Retrospective	62	30.2	61 M, 1 F	19% smokers	62 cases of myocarditis	1		Supportive	R	Y,Pt1: no clear cellular infiltrate, Pt2: mild lymphocytic infiltrate; Pt3: eosinophilic infiltrate
			16		11 M, 5 F	NR	ACS:16	1		Supportive	R	
Cangemi et al., 1958 [47]	USA	Case report	1	56	M	NR	1 pericarditis	1	16	Supportive	R	
Taylor et al., 2012 [48]	USA	Case report	1	32	M	Smoker	1 myopericarditis	1	14	Aspirin, morphine, nitroglycerin	R	N
Murphy et al., 2003 [49]	USA	Case report	1	29	M	NR	1 myocarditis	1	21	GDMT for HF; prednisone	R	Y,mixed eosinophilic-lymphocytic
Guerdan et al., 2004 [50]	USA	Case report	1	26	M	NR	1 myocarditis	1	11	NSAIDS, and pain control	R	N
Docekal et al., 2019 [51]	USA	Case report	1	20	M	NR	Recurrent epicardial V. tach	1	28	EP ablations were attempted.	R	N
Jones et al., 1964 [52]	USA	Case report	1	39	M	NR	1 myocarditis	1	11	Supportive	D	Y,mixed infiltrate of mononuclear cells and edematous IV septum and ventricles; necrotic foci containing eosinphils
Dalgaard et al., 1954 [53]	Norway	Case report	1	22	M	NR	1 myocarditis	Revaccine	8	Supportive	D	Y,Autopsy: Numerous foci of acute degeneration with loss of transverse striation, granular necrosis of myofibrils, and pronounced infiltration with granulocytes and lymphocytes. Between these areas normal myofibrils were found.
Bruner et al., 2014 [54]	USA	Case report	1	27	M	NR	1 myopericarditis	1	14	Thrombolysis	R	N

TTC, takotsubo cardiomyopathy; IHD, Ischemic heart disease; HLD, hyperlipidemia; NSAID, nonsteroidal anti-inflammatory drug; HTN, hypertension; COPD, chronic obstructive pulmonary disease; CKD, chronic kidney disease; DM, diabetes mellitus, ECMO, extracorporeal membrane oxygenation; STEMI, ST elevation myocardial infarction; CAD, coronary artery disease; RCA, right coronary artery; GDMT: Guideline directed medical therapy; HF: heart failure; EP: electrophysiology; R, recovery; D, death; M, male; F, female; EMB, endomyocardial biopsy; NR, Not reported; pts, patients.

**Table 5 vaccines-10-00700-t005:** Adverse events after other vaccinations.

Author, Year	Country	Type of Study	Number of Cases	Median Age	Gender	Patient Comorbidities	Event	Dose after Which Event Happened	Days after Vaccination Symptoms	Treatment	Outcome	EMB
**Pneumococcal vaccination**												
Makaryus et al., 2006 [56]	USA	Case report	1	71	F	NR	Myocarditis	NA	Supportive: GDMT for HF	R	No	NA
Tawfik et al., 2017 [57]	USA	Case report	1	75	F	NR	Pericardial effusion	18	Prednisone for 2 weeks tapered over 3 months	R	Yes; organized effusion containing hemorrhage, acute inflammation with reactive changes and mesothelial hyperplasia	NA
**Tdap vaccination**												
Boccara et al., 2001 [58]	USA	Case report	1	31	M	NR	Myopericarditis	Re vaccine	NA	Aspirin	R	Y; arteriolar smooth muscle contraction was prominent with increased permeability demonstrated by interstitial edema and diapedesis of erythrocytes.
Clark et al., 2014 [60]	USA	Case Report	1	28	M	Smoker	Myopericarditis	Re vaccine	3 days	Colchicine and DAPT	R	N
**Tetanus Toxoid**												
Yamamoto et al., 2018 [59]	Japan	Case report	1	18	M	NR	Myocarditis	Re-vaccine	NA	Steroids	R	Yes: perivascular eosinophilic infiltrates with myocyte necrosis and abundant interstitial lymphocytic infiltrates with myocyte necrosis, separately. Numerous eosinophilic infiltrations with degranulating eosinophils were observed in the perivascular regions. Perivascular interstitial fibrosis was also observed Immunostaining for eosinophilic cationic protein showed extensive staining in the myocardial interstitium.
**Cholera vaccine**												
Koutsaimanis et al., 1978 [61]	England	Case Report	1	40	M	NR	Anterior wall MI	1	6 days	NA	R	N
**Rabies vaccine**												
Lyon et al., 1948 [62]	Jerusalem	Case Report	1	58	M	NR	Myocarditis	1	Few weeks after 14 injections	VItamin B complex and sedrena extract	R	N

MI, myocardial infarction; CAD, coronary artery disease; GDMT: guideline directed medical therapy; HF: heart failure; EP: electrophysiology; R, recovery; D, death; M, male; F, female; DAPT: Dual antiplatelet therapy; EMB, endomyocardial biopsy; NR, not reported.

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
