# Peer review of "Cardiac Adverse Events after Vaccination—A Systematic Review"

_vaccines, 2022, doi:10.3390/vaccines10050700_

Round 1

Reviewer 1 Report

The title should go like “Cardiac Adverse Events after Vaccination-A Systematic Review

Line 11 …… please replace “following several vaccines” with “after COVID-19 Vaccination.”

Line 13 please be specific about the “Inception”

Line 13 A total of 52 studies were………what criteria were used to select these studies? Please mention   

Line 14 to 16 these sentences could be merged

Line 19 please close the space in between “event. Adverse”

Line 38 to 39 Wuhan, China

Line 29 please update the global infection and death rates of covid 19

Line 46 which studies? Please mention and highlight those studies here.

Line 61 why were studies involving patients<18 years of age excluded? Why did you include English as criteria? Or did you use any text in foreign languages?  

Line 74 please include “The PRISMA flow diagram for systematic review”

Line 76 ….7327 studies were found on PubMed not “in”

Line 80 to 84 please this could be written better. Please improve

Line 85 43.79±21.2 years

Line 88 please refer to the comment above

Table 1 please use number citation as in the text do same for table 2, 3, 4. Also Patient comorbities should be not reported (NR). What volume of dose was given. Please mention do same for all tables

Line 110 please unbold

Author Response

The title should go like “Cardiac Adverse Events after Vaccination-A Systematic Review

Changed

Line 11 …… please replace “following several vaccines” with “after COVID-19 Vaccination.”

Reworded

VAERS has been used to report adverse events following several vaccines and not just COVID-19 vaccine. Therefore the use of word “several” vaccine. Language is reworded to make it more clear

Line 13 please be specific about the “Inception” - Date?

First study which we found included was April 1947

However, during the search strategy no beginning date was specified and all data was picked up based on earliest data the search engine could pick up.

Line 13 A total of 52 studies were………what criteria were used to select these studies? Please mention   

Changed.

We searched the electronic databases PubMed, EMBASE and Scopus from inception till July 9, 2021, for any study describing cardiac adverse event attributed to the vaccination. A total of 52 studies met the criteria with a total of 324 patients

Line 14 to 16 these sentences could be merged

Done

Line 19 please close the space in between “event. Adverse”

Changed

Line 38 to 39 Wuhan, China

Changed

Line 29 please update the global infection and death rates of covid 19

It rapidly spread, resulting in a global pandemic affecting 400 million people worldwide and has taken approximately 6 million lives. In order to fight this infection there was an emergent authorization of the vaccines.

Line 46 which studies? Please mention and highlight those studies here.

Changed

Line 61 why were studies involving patients<18 years of age excluded? Why did you include English as criteria? Or did you use any text in foreign languages?  

We expected separate pathophysiology and management in paediatric population.

Yes, we did find studies in other languages however they were excluded because of the authors unfamiliarity with other languages

Line 74 please include “The PRISMA flow diagram for systematic review”

Included

Line 76 ….7327 studies were found on PubMed not “in”

Changed

Line 80 to 84 please this could be written better. Please improve

Changed

Line 85 43.79±21.2 years

Changed

Line 88 please refer to the comment above

Changed

Table 1 please use number citation as in the text do same for table 2, 3, 4. Also Patient comorbidities should be not reported (NR). What volume of dose was given. Please mention do same for all tables.

Changed to NR
Volume of dose not consistently reported therefore not included.

Line 110 please unbold

Done

Reviewer 2 Report

In this manuscript the authors present a literature search on the emergence of cardiac disease, in particular myocarditis and pericarditis, in patients after vaccinations for various targets using various types of vaccins. Out of 7327 studies extracted from different databases, 267 reports qualified for further investigation: finally 52 studies were selected, with a total of 324 patients. These patients are presented as case reports in different tables, categorized according to target of vaccination: these targets included different types of vaccins. Most patients were in the age category between 40 and 60 years, without a clear difference between males and females. The cardiac features were variable, but most shared inflammatory responses, that received successful treatment with anti-inflammatory drugs. The mortality was very low. There appeared no unequivocal causal relationship between the vaccination and emergence of cardiac adverse side effect.

This is an interesting review. The design and data presentation is clear and the discussion is well written. This is a relevant topic in view of all discussions in the public about hesitation to vaccination claiming the importance of adverse side effects in unwillingness to receive a vaccination. The authors did not find a solid rationale for this unwillingness to get vaccinated for major vaccines used to day in the area of COVID-19 (irrespective of type of vaccine), and also influenza vaccines, tetanus toxoid, smallpox vaccines, pneumococcal, and cholera vaccination and rabies vaccination.

There are a few minor point that can improve the quality of this report:

  • It is advised to restructure the tables so that these become better readable
  • It is advised to add a summary table with the most relevant data, to increase the take-home message
  • It is advised to keep abbreviations to a minimum, especially in the text
  • Figure 1 is lacking in the manuscript
  • It is advised to rephrase the discussion. The discussion should have statements regarding the causality of cardiac adverse events and (type/target of) vaccination; the severity and mortality of adverse side effects in relation to occurrence in non-vaccinated individuals; and comment on safety of vaccination in relation to cardiac adverse side effects. Also it can be stated that any vaccine hesitancy cannot be based on expectations regarding cardiac side effects. The conclusion regarding further research may have no sense as the points raised for further investigation cannot simply be addressed considering the low numbers of patients affected, and the heterogeneity in expression.

Author Response

Thanks for the thoughtful comments and your time in reviewing this manuscript. 

1.It is advised to restructure the tables so that these become better readable

We tried to put tables in better layout. However, if reviewer wants, we can put all the table in supplementary. And keep the summary table in the manuscript if that makes it more readable. Any suggestions are welcome.

2 It is advised to add a summary table with the most relevant data, to increase the take-home message

Absolutely agreed. I think it will be a good idea to summarize the tables given the complexity of the data presented. Added

3.It is advised to keep abbreviations to a minimum, especially in the text

Agree. Removed some

4.Figure 1 is lacking in the manuscript

Added

5.It is advised to rephrase the discussion. The discussion should have statements regarding the causality of cardiac adverse events and (type/target of) vaccination; the severity and mortality of adverse side effects in relation to occurrence in non-vaccinated individuals; and comment on safety of vaccination in relation to cardiac adverse side effects.

Compared it with viral myocarditis where incidence rate is much higher. Added to the manuscript.

Also compared incidence between vaccinated and unvaccinated.

Also, it can be stated that any vaccine hesitancy cannot be based on expectations regarding cardiac side effects.

Agreed. Stated

The conclusion regarding further research may have no sense as the points raised for further investigation cannot simply be addressed considering the low numbers of patients affected, and the heterogeneity in expression.

Agree with the reviewer. Removed the statement about further research as low number of patients limit any translational research.

Reviewer 3 Report

Cardiac adverse events are reported following several vaccines. In the paper, the authors have searched the electronic databases PubMed, EMBASE and Scopus untill July 9, 2021, retrieving 52 studies with a total of 324 patients. They found 10 studies describing adverse events after influenza vaccination. 18 studies described adverse events after COVID-19 vaccination,3 studies with tetanus toxoid, and 1 study with cholera vaccination and 1 study with rabies vaccine. Pericardium involvement was the most reported adverse event for influenza vaccine. Adverse events following COVID-19 vaccination happened at mean age of 42.7 years, more common in males and more likely after second dose. Acute coronary syndrome has been seen with some vaccines.

The paper is well written and covers a hot topic, which may arise the interest of readers belonging to the academia and the physicians. The Results are well described. The authors have also tried to explain some molecular mechanisms underlying the development of the cardiac adverse effects.

Suggestions for improvement

-The authors could highlight better the limitations of the study. In particular, from the systematic review of the reports covered in the present manuscript, it is not possible to determine the relative risk of developing cardiac adverse events following any of the vaccination in current use. This could be better stated in the Discussion and Conclusion sections.

-A few misspellings and typos should be revised

Recommendation

Overall, the paper deserves to be published in Vaccines.

Author Response

Thanks for the thoughtful comments and your time in reviewing this manuscript. 

-The authors could highlight better the limitations of the study. In particular, from the systematic review of the reports covered in the present manuscript, it is not possible to determine the relative risk of developing cardiac adverse events following any of the vaccination in current use. This could be better stated in the Discussion and Conclusion sections.

Limitations added separately.

-A few misspellings and typos should be revised

Revised

Round 2

Reviewer 1 Report

The authors have revised the manuscript and it read well now. I, therefore, recommend accepting in the present form